# Relationship between Serum Ferritin Level and Dyslipidemia in US Adults Based on Data from the National Health and Nutrition Examination Surveys 2017 to 2020

**DOI:** 10.3390/nu15081878

**Published:** 2023-04-13

**Authors:** Guohua Li, Wenlu Yu, Hexiang Yang, Xinyue Wang, Tianyou Ma, Xiaoqin Luo

**Affiliations:** Department of Nutrition and Food Safety, School of Public Health, Xi’an Jiaotong University, Xi’an 710061, China

**Keywords:** serum ferritin, dyslipidemia, iron intake, TC, LDL-C, TG, HDL-C

## Abstract

Previous research has suggested that high serum ferritin (SF) levels may be associated with dyslipidemia. This study investigated the association between SF levels and dyslipidemia in American adults, which held relevance for both clinical and public health areas concerned with screening and prevention. Data from the pre-pandemic National Health and Nutrition Examination Surveys (NHANES), conducted between 2017 and 2020, were utilized for this analysis. Multivariate linear regression models were used to explore the correlation between lipid and SF concentrations, and the connection between SF and the four types of dyslipidemia was further assessed by using multivariate logistic regression analysis. Odds ratios (ORs; 95% CI) for dyslipidemia were calculated for quartiles of SF concentrations, with the lowest ferritin quartile as the reference. The final subjects consisted of 2676 participants (1290 males and 1386 females). ORs for dyslipidemia were the highest in the fourth quartile (Q4) of SF both in males (OR: 1.60, 95% CI: 1.12–2.28) and females (OR: 1.52, 95% CI: 1.07–2.17). The crude ORs (95% CI) for the risk of High TC and High LDL-C increased progressively in both genders. However, after adjusting for covariates, the trend of significance was only present in females. Finally, the association between total daily iron intake and the four types of dyslipidemia was examined, revealing that the risk of High TG in the third quartile of the total daily iron intake was 2.16 times greater in females (adjusted OR: 3.16, 95% CI: 1.38–7.23). SF concentrations were remarkably associated with dyslipidemia. In females, daily dietary iron intake was associated with High-TG dyslipidemia.

## 1. Introduction

Dyslipidemia, characterized by elevated total cholesterol (TC), low-density lipoprotein (LDL-C), and triglycerides (TGs) or reduced high-density lipoprotein cholesterol (HDL-C) is a prevalent condition of the abnormal metabolism of lipoproteins in the human body [1,2]. The onset of dyslipidemia is insidious with no obvious symptoms in the early stages, but it frequently contributes to serious cardiovascular diseases such as coronary heart disease and stroke [3]. Cardiovascular disease represents a leading global cause of morbidity and mortality [4], and increased concentrations of TC or LDL-C, HDL-C, and TC/HDL-C, which can impair endothelial function [5], are strong predictors for coronary heart disease [6]. Moreover, multiple studies have demonstrated that dyslipidemia is associated with obesity, hypertension, Type 2 diabetes mellitus, and numerous other diseases [7,8,9]. Consequently, the prevention and control of dyslipidemia can help in promoting the primary and secondary prevention of many severe diseases including cardiovascular events.

The majority (80%) of dyslipidemias are triggered by the diet and lifestyle factors [10], such as sedentary behavior, and the increased intake of high saturated fats and refined sugars [11,12]. Both lifestyle modifications and pharmacological interventions can improve lipid profiles [13], and anthropometric indices provide predictive values for dyslipidemia [14]. However, no studies are available to identify biochemical indicators that are predictive of dyslipidemia in humans.

Iron, an essential micronutrient for human health, can exert potentially toxic effects on cells. Due to its involvement in pathological catalytic or oxidative stress reactions, excessive human exposure to iron may induce disease [15]. Following absorption through the small intestine, iron reaches the iron-demanding tissues via transferrin, and the remaining iron can then be stored as ferritin. Serum ferritin (SF) serves as a clinical biomarker for evaluating the iron status and is crucial in regulating iron homeostasis [16,17]. Serum ferritin is an acute-phase protein that is elevated in conditions such as inflammation or tissue damage [18]. Some research has suggested that dyslipidemia and unfavorable lipid ratios may be relevant to an elevated SF level [19]. High iron levels can disturb the cholesterol levels by inducing insulin resistance, interfering with lipid oxidative stress, and raising proinflammatory cytokines and other mechanisms, thereby promoting cardiovascular disease [20,21,22,23]. Obesity, metabolic syndrome, atherosclerosis, diabetes mellitus, dyslipidemia, and cancer were also associated with excessive SF levels [24,25,26,27,28]. In addition, ferritin levels are influenced by dietary iron intake, with some studies indicating that a higher total dietary iron level may correlate with an increased risk of cardiovascular disease and cancer mortality [29].

Limited research has investigated the association between SF and dyslipidemia, primarily focusing on specific populations such as adolescents or diabetics [30,31], with inconsistent findings. Additionally, few studies on SF and dyslipidemia have considered total dietary iron intake. To comprehensively evaluate this complex correlation, large-scale epidemiological investigations are warranted. National Health and Nutrition Examination Survey (NHANES) is a nationally representative project conducted by the Centers for Disease Control and Prevention’s National Center for Health Statistics [32]. This program assesses the health and nutritional status of adults and children in the United States through interviews and physical examinations with precise specifications and operational requirements [33]. Utilizing data from the NHANES cycle between March 2017 and pre-pandemic 2020, this study examines whether ferritin could serve as a predictor of dyslipidemia, and the relationship between daily dietary iron intake and dyslipidemia in US adults.

## 2. Materials and Methods

### 2.1. Data Source and Study Population

NHANES is a program of studies designed to assess the health and nutritional status of adults and children in the United States. The survey combines interviews and physical examinations [33]. This study included participants from the NHANES cycle between March 2017 and the pre-pandemic period in 2020. A nationally representative sample of 15,560 participants was recruited. Individuals without SF or serum lipid data (TC, LDL-C, TG, and HDL-C), those younger than 20 years old, in pregnancy or lactation, and taking lipid-modifying medications were excluded. Ultimately, 2676 subjects were included in the data analysis. The detailed elimination process is described in Figure 1.

### 2.2. Serum Ferritin and Blood Lipids

The serum ferritin and blood lipid values which we utilized in this study were derived from the existing NHANES datasets already included in NHANES. Serum specimens underwent processing, storage, and shipment to the Division of Laboratory Sciences, National Center for Environmental Health, Centers for Disease Control and Prevention, Atlanta, GA, for analysis [34]. The cobas^®^ e601 system was used for serum ferritin measurement, employing a sandwich principle with a total duration time of 18 min. Both fresh and frozen serum or plasma samples were suitable for ferritin analysis. Samples were stored at temperatures below −20 °C until analyzed, and for long-term storage, specimens were frozen at −70 °C. The lower detection limit for SF was 0.5 ng/mL.

For lipid measurements, serum specimens were processed, stored, and shipped to the University of Minnesota, Minneapolis, MN [34]. An enzymatic assay was employed for total cholesterol measurement. Serum TG and HDL-C were measured photometrically. The Friedewald equation was used to calculate LDL-C from direct measurements of TC, TG, and HDL-C.

### 2.3. Definition of Dyslipidemia

According to the Third Report of the National Cholesterol Education Program (NCEP Expert Panel on Detection, Evaluation, and Treatment of High Blood Cholesterol in Adults (Adult Treatment Panel III)) [35], the diagnosis of dyslipidemia was assigned if the participant met any of the following criteria: (1) High TC: TC ≥ 6.21 mmol/L (240 mg/dL); (2) High TG: TG ≥ 2.26 mmol/L (200 mg/dL); (3) High LDL-C: LDL-C ≥ 4.14 mmol/L (160 mg/dL); (4) Low HDL-C: for males < 1.03 mmol/L (40 mg/dL), for females < 1.29 mmol/L (50 mg/dL); (5) the patient was taking lipid-regulating drugs.

### 2.4. Assessment of Covariates

The NHANES included information on the socio-demographic and lifestyle characteristics through interviews administered by trained interviewers. Some studies have reported significant differences in the prevalence of dyslipidemia by age or race/ethnicity [36,37]. We categorized participants by age as young adults (20–44 years), middle-aged adults (45–64 years), and older adults ≥ 65 years according to the US population definition. Based on the NHANES sampled population, we defined race/ethnicity as Mexican American, Non-Hispanic white, Non-Hispanic Black, Non-Hispanic Asian, and Others. Statistical analytic models also included a six-month time period (1 November through 30 April, 1 May through 31 October), education level (High school and below, Above high school), marital status (Married/Living with Partner, Widowed/Divorced/Separated, Never married). Alcohol consumption status was based on whether participants had consumed at least 12 alcoholic drinks per year or not. Participants were considered smokers if they had smoked ≥100 cigarettes in their life. Blood pressure questionnaire was administered by trained interviewers using the Computer-Assisted Personal Interview system. Body Mass Index (BMI) was calculated based on height and weight. Dietary intake data were collected in person at the Mobile Examination Center. The data primarily assess the types and amounts of food and beverages (including all kinds of water) consumed by subjects in the 24 h (midnight to midnight) before the interview and estimated the total energy, nutrient value, and other components consumed from these foods and beverages.

### 2.5. Statistical Analyses

Quantitative data were summarized as means ± standard deviations (SD) for normal variables and as medians (interquartile ranges) for skewed variables. Qualitative data were presented as a percentage. The *t*-test or Kruskal–Wallis test was used to analyze the relationship between continuous variables. The chi-square test was used for categorical variables.

Multivariable linear regression models were employed to investigate the correlation between blood lipid (TC, LDL-C, TG, and HDL-C) and SF concentrations, which were naturally log-transformed due to skewed distribution. The resulting regression coefficients were back-transformed and presented as percent differences.

SF levels were divided into quartiles based on the distribution in different-genders participants: for males, those were ≤103.00, 103.00–164.50, 164.50–263.75, and ≥263.75 μg/L; for females, ≤32.23, 32.23–64.35, 64.35–120.75 and ≥120.75 μg/L. The 1st quartile (Q1) and 4th quartile (Q4) represented the lowest and highest ferritin levels, respectively.

There was a gender difference in serum ferritin and blood lipid concentrations [38]. We used a gender-stratified analysis to explore the relationship between serum ferritin and blood lipids. Logistic regression analysis was performed to estimate the odds ratio (ORs) and 95% confidence intervals (95% CIs) of the association between SF and daily total iron intakes and risk of the 4 types of dyslipidemia (High-TC, High-LDL cholesterol, High-TG, Low-HDL cholesterol). The Q1 of SF was used as the reference group in stratified analysis by gender (male and female). The crude model examined the association between SF and the 4 types of dyslipidemia without adjustment for any covariates. The adjusted model included demographic characteristics. All the analyses were two-sided, and *p* < 0.05 was considered statistically significant. All analyses and Figure 2 and Figure 3 were performed with R 12.0.

## 3. Results

### 3.1. Basic Characteristics of Individuals by Gender

The basic characteristics of all participants are presented in Table 1. There were 1290 (48.2%) males and 1386 (51.8%) females. Education levels, marital status, drinking and smoking status, and BMI showed significantly different between the genders. Age and race/ethnicity distributions were comparable between males and females. The prevalence of hypertension was approximately 30% in both males and females. Males have higher SF levels (164.50 μg/L and 64.35 μg/L), greater total energy (2349.50 kcal and 1731.00 kcal), and higher total fat intakes (90.28 g and 68.09 g) than females, and the differences were statistically significant. Regarding lipid levels, males were more likely to have higher LDL-C (2.90 mmol/L), TG (1.04 mmol/L), and lower HDL-C (1.22 mmol/L) than females.

### 3.2. Association between Serum Ferritin and Lipid Levels

Multivariable adjusted linear regression results showed positive associations between blood TC as well as TG and SF concentrations in both males and females (all *p* < 0.05). In females, SF level was also positively associated with LDL-C levels, whereas it was marginally negatively associated with the HDL-C level (*p* = 0.070), although no statistical significance was obtained (Table 2).

### 3.3. Association between Quartile of Serum Ferritin Level and Dyslipidemia

We further examined the gender-specific correlations of SF with dyslipidemia, and the comparative ORs (95% CIs) of the SF quartile categories are shown in Figure 2. A remarkable trend of increasing risk of dyslipidemia was observed in men with increasing SF levels (*p* for trend = 0.005), even after adjusting possible covariates (*p* for trend = 0.019). Similarly, an increased SF trended towards marginally significantly increases in the likelihood of dyslipidemia in females (*p* for trend = 0.072) even though the association was attenuated after controlling for covariates (*p* for trend = 0.104). Regardless of gender, the multivariable-adjusted ORs of Q4 were the highest, 1.60, 95% CI: 1.12, 2.28 and 1.52, 95% CI: 1.07, 2.17, respectively (all *p* < 0.01).

### 3.4. Association between Serum Ferritin Quartiles and Four Specific Types of Dyslipidemia

The ORs (95% CIs) of the four specific types of dyslipidemia according to the categories of SF by gender is shown in Figure 3. The Q4 crude ORs (95% CIs) for the risk of High TC, High LDL-C, and High TG levels were the highest in males. The crude ORs (95% CIs) for the risk of High TC and High LDL-C levels increased progressively across the SF quartiles in females (all *p* < 0.001 for the trend) and in males (all *p* < 0.05 for the trend). However, after adjusting for covariates, the significant trends only persisted in females. The Q2, Q3, and Q4 adjusted ORs (95% CIs) for High TC and High LDL-C in females were 2.78 (95% CI: 1.37–5.67), 3.29 (95% CI: 1.636.63), 3.97 (95% CI:1.97–7.97), and 2.70 (95% CI: 1.31–5.55), 3.00 (95% CI: 1.48–6.1), 3.24 (95% CI: 1.59–6.59), respectively.

### 3.5. Association between Quartile of Total Iron Intake and Dyslipidemia by Gender

Ferritin is a form of iron stored in the body and is influenced by dietary iron intake. Therefore, we also explored the relationship between daily total iron intake and four specific types of dyslipidemia. No association between daily total iron intake and any kind of dyslipidemia was detected in males. Interestingly, for females, there was a positive association between the Q3 of daily total iron intake and High TG both in the crude model and adjusted model (Table 3), respectively (crude OR: 2.60, 95% CI: 1.27–5.34; adjusted OR: 3.16, 95% CI: 1.38–7.23).

## 4. Discussion

The results of this study indicate that SF levels are positively correlated with increased levels of TC, LDL-C, and TG, but negatively correlated with increased HDL-C levels in US adults. Additionally, individuals with the Q4 of SF exhibited the highest risk of dyslipidemia. In the analysis of SF and the four types of dyslipidemia, we observed gender differences. The trends in the risk High TC, High LDL-C, and High TG varied with increasing SF quartile in the males and females. In exploring the involvement of diets, we determined that females with higher daily iron intake, which may primarily reflect the SF levels, had an elevated risk of High-TG dyslipidemia, with the highest risk observed in Q3.

Previous studies have demonstrated that elevated SF is associated with the risk of cardiovascular disease [39,40,41]. Dyslipidemia is estimated to account for over one-third of deaths resulting from ischemic heart disease or ischemic stroke in both developed and developing countries [42]. Consequently, we investigated the potential of SF as a preventive indicator for dyslipidemia. Our findings suggest that higher SF levels are more likely to be associated with dyslipidemia in American adults, consistent with studies from China [23] and the Middle East. The latter showed a progressively significant increase dyslipidemia risk from Q1 to Q4, a twofold increase in the risk at Q4 compared to Q1 in males and females [43]. However, regarding specific types of dyslipidemia, we observed gender discrepancies. While positive associations existed between blood TC as well as LDL-C and SF concentrations for both males and females, a significantly increasing trend for High-TC and High-LDL-C dyslipidemia was observed only in females after adjustment. On the contrary, Kim et al. reported that TC, LDL-C, and TG levels in Korean adolescents were significantly correlated with SF in boys but not in girls [44]. This discrepancy may be age-related as iron storage and metabolism in adolescents differ from those of adults, especially with postmenopausal women [45]. One study illustrated that TG dyslipidemia was more prevalent in individuals with the highest compared to the lowest SF levels among postmenopausal women and men [46]. These results were not observed in females in our study possibly due to the inclusion of non-menopausal women. A previous study identified strong gender differences in associations between ferritin and MetS risk (including diabetes and dyslipidemia). This indicated that elevated concentrations of ferritin were associated with a higher risk of dyslipidemia, obesity, overweight, and diabetes in men, while there was no such association in women [38]. Factors such as gender-linked gene expression, metabolic control, sex hormones, and environmental lifestyle may contribute to these differences. In adults, men typically have more visceral fat accumulation and lower plasma adiponectin levels, which may lead to gender differences in vulnerability to cardiovascular disease. Further investigation of the mechanisms behind gender differences is critical.

It is well established that SF is a marker reflecting the body’s iron stores and an acute-phase protein, influenced by dietary iron intake [47]. Thus, we extended our observations to explore the relationship between total daily dietary iron intake and dyslipidemia. Our findings revealed that the risk of High-TG dyslipidemia was highest in the Q3 of dietary iron intake in females, suggesting that higher iron intake is associated with a higher risk of High TG. However, the same results were not observed in males, and the exact reason for it was unclear. This outcome is inconsistent with the relationship between SF and dyslipidemia, making it challenging to determine whether dyslipidemia is caused by elevated iron storage or serum ferritin acting as an inflammatory marker. Further prospective research is necessary to verify these findings. Currently, another study indicated that higher heme iron intake is positively associated with elevated TG levels [48]. We did not conduct further studies on the relationship between heme and non-heme dietary iron and dyslipidemia due to the lack of dietary data.

The mechanism underlying the association between SF levels and dyslipidemia remains uncertain. Serum ferritin overload responds to iron overload. Iron can cause oxidative stress-induced tissue damage by catalyzing the conversion of hydrogen peroxide to free radicals. Chronic oxidative stress is linked to damage to mitochondrial β-oxidation of long-chain fatty acids in the pancreas. Some studies propose a connection to the inflammatory response. Iron overload was significantly associated with oxidative stress markers (MDA and TAC), liver damage, lipid distribution, and other variables [49]. Evidence from animal studies suggests complex interactions between inflammatory cytokines and iron in regulating secretion [50]. Therefore, prospective studies are needed to confirm the mechanisms driving the relationship between SF and dyslipidemia. Furthermore, follow-up studies should determine whether elevated iron stores precede dyslipidemia and lead to an increased risk associated with dyslipidemia and cardiovascular disease. A larger sample size is required to establish the range of serum ferritin within which dyslipidemia occurs.

The significant strength of our study was that we conducted a survey based on a stratified, multi-stage probability sampling design using nationally representative data. In addition, we examined the potential role of daily total iron intake in this relationship. The present study also has several limitations. Firstly, the data were not adjusted for the active inflammation-related marker (C-reactive protein) due to missing data, but we initially excluded patients with a history of acute inflammation and Chen et al. reported no statistically significant effect of inflammatory factors on the correlation between SF and Metabolic syndrome [51]. Secondly, the pregnancy status and menstruation of some females were unknown, potentially introducing bias to the results. Lastly, the dietary iron intake data were not adjusted by energy intake, as the majority of subjects’ intake data of three macronutrients (lipid, protein, and carbohydrate) were unavailable.

In conclusion, our research discovered that SF levels were significantly associated with dyslipidemia. However, gender differences in the results of SF were associated with the four types of dyslipidemia. Daily dietary iron intake was associated with High-TG dyslipidemia in females. The mechanism of the relationship between SF and dyslipidemia remains unexplained, but SF may be a suitable biomarker for dyslipidemia.

## Figures and Tables

**Figure 1 nutrients-15-01878-f001:**
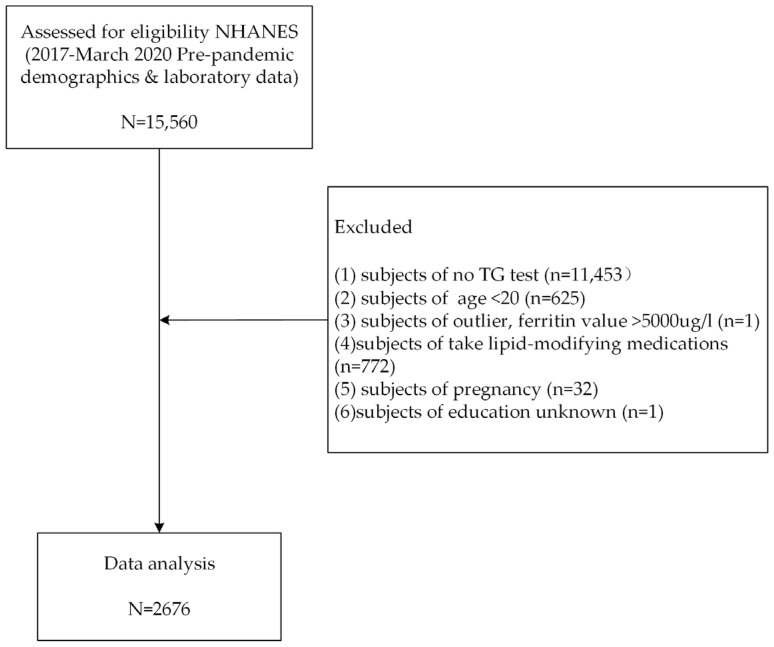
Flowchart showing the selection of participants in the study.

**Figure 2 nutrients-15-01878-f002:**
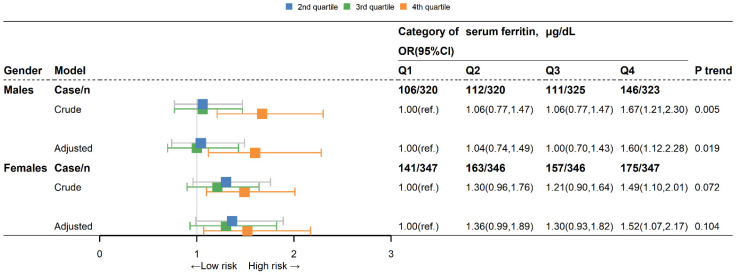
ORs (95% CIs) for dyslipidemia according to serum ferritin in males and females. OR, odds ratio; CI, confidential intervals; Q1: 1st quartile, Q2: 2nd quartile, Q3: 3rd quartile, Q4: 4th quartile; TC, total cholesterol; LDL-C, low-density lipoprotein cholesterol; TG, triglyceride; HDL-C, high-density lipoprotein cholesterol. OR and 95% CI were obtained by multivariable logistic regression analysis (crude models). The adjusted models were adjusted for age, race, six-month time period, educational level, marital status, BMI, hypertension, smoking status, alcohol status, total daily energy intake, and total daily fat intake.

**Figure 3 nutrients-15-01878-f003:**
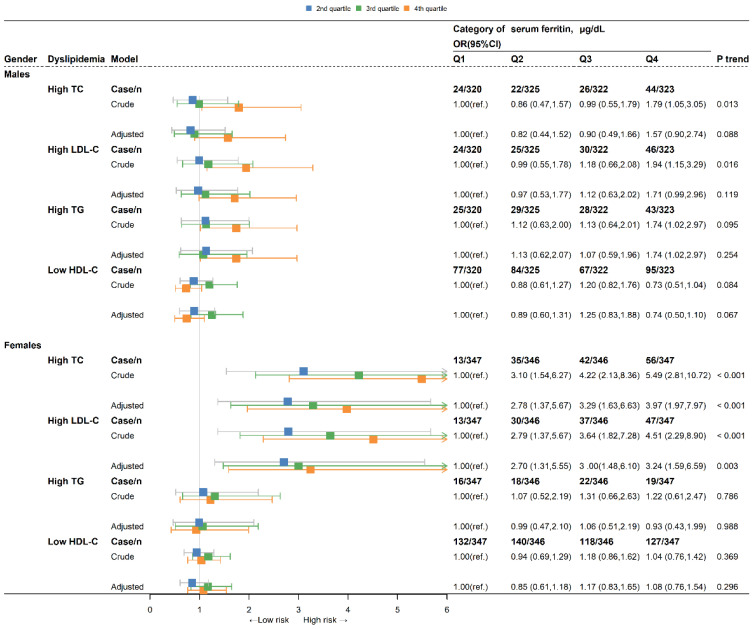
ORs (95% CIs) for four types of dyslipidemia according to serum ferritin in males and females. OR, odds ratio; CI, confidential intervals; Q1: 1st quartile, Q2: 2nd quartile, Q3: 3rd quartile, Q4: 4th quartile; TC, total cholesterol; LDL-C, low-density lipoprotein cholesterol; TG, triglyceride; HDL-C, high-density lipoprotein cholesterol. OR and 95% CI were obtained by multivariable logistic regression analysis (crude models). The adjusted models were adjusted for age, race, six-month time period, educational level, marital status, BMI, hypertension, smoking status, alcohol status, total daily energy intake, and total daily fat intake.

**Table 1 nutrients-15-01878-t001:** Baseline characteristics of the subjects according to gender.

Characteristics	Total (%)	Gender		*p* Value
	*n* = 2676	Males (%)	Females (%)	
		*n* = 1290 (48.2)	*n* = 1386 (51.8)	
Age				0.246
20–44	1257 (47.0)	603 (46.7)	654 (47.2)	
45–64	984 (36.8)	462 (35.8)	522 (37.7)	
≥65	435 (16.3)	225 (17.4)	210 (15.2)	
Race/Ethnicity (%)				0.365
Mexican American	352 (13.2)	177 (13.7)	175 (12.6)	
Non-Hispanic White	880 (32.9)	440 (34.1)	440 (31.7)	
Non-Hispanic Black	693 (25.9)	331 (25.7)	362 (26.1)	
Non-Hispanic Asian	332 (12.4)	147 (11.4)	185 (13.3)	
Others	419 (15.7)	195 (15.1)	224 (16.2)	
Six-month time period				0.806
1 November through 30 April	1412(52.8)	677(52.5)	735(53.0)	
1 May through 31 October	1264(47.2)	613(47.5)	651(47.0)	
Education level (%)				0.004
High school and below	1088 (40.7)	562 (43.6)	526 (38.0)	
Above high school	1588 (59.3)	728 (56.4)	860 (62.0)	
Marital status (%)				<0.001
Married/Living with Partner	1534 (57.3)	782 (60.6)	752 (54.3)	
Widowed/Divorced/Separated	530 (19.8)	214 (16.6)	316 (22.8)	
Never married	612 (22.9)	294 (22.8)	318 (22.9)	
Drinking status (%)				<0.001
Yes	2446 (91.4)	1220 (94.6)	1226 (88.5)	
No	230 (8.60)	70 (5.40)	160 (11.5)	
Smoking status (%)				<0.001
Yes	1095 (40.9)	648 (50.2)	447 (32.3)	
No	1581 (59.1)	642 (49.8)	939 (67.7)	
BMI *	29.57 ± 7.49	28.75 ± 6.30	30.33 ± 8.37	<0.001
Hypertension (%)				0.536
Yes	792 (29.6)	374 (29.0)	418 (30.2)	
No	1884 (70.4)	916 (71.0)	968 (69.8)	
Daily total energy intake (kcal)	1989.00 (1431.00, 2694.00)	2349.50 (1718.25, 3084.00)	1731.00 (1273.00, 2290.50)	<0.001
Daily total fat intake (gm)	77.32 (53.44, 112.03)	90.28 (60.41, 127.38)	68.09 (47.97, 96.80)	<0.001
SF ^ (μg/L)	106.00 (51.05, 199.25)	164.50 (103.00, 263.75)	64.35 (32.23, 120.75)	<0.001
TC ^ (mmol/L)	4.76 (4.14, 5.48)	4.73 (4.11, 5.40)	4.81 (4.16, 5.51)	0.114
LDL-C ^ (mmol/L)	2.87 (2.30, 3.49)	2.90 (2.33, 3.52)	2.85 (2.28, 3.44)	0.042
TG ^ (mmol/L)	0.95 (0.66, 1.41)	1.04 (0.69, 1.53)	0.88 (0.63, 1.31)	<0.001
HDL-C ^ (mmol/L)	1.32 (1.11, 1.58)	1.22 (1.04, 1.47)	1.42 (1.22, 1.71)	<0.001

* Values are presented as mean ± SD; ^ Values are presented as median (interquartile range); BMI, body mass index; SF, serum ferritin; TC, total cholesterol; LDL-C, low-density lipoprotein cholesterol; TG, triglyceride; HDL-C, high-density lipoprotein cholesterol.

**Table 2 nutrients-15-01878-t002:** Adjusted percent difference (%) and 95% CI in serum ferritin measures in relation to lipid concentrations.

Blood Lipid% ^1^ (mmol/L)	Serum Ferritin (95% CI)			
	Males	*p*-Value	Females	*p*-Value
TC	1.65 (0.05, 3.29)	0.044	1.63 (0.39, 2.87)	0.010
LDL-C	1.66 (−0.84, 4.22)	0.195	2.72 (0.78, 4.70)	0.006
TG	5.69 (1.48, 10.07)	0.008	3.51 (0.49, 6.62)	0.022
HDL-C	−0.13 (−1.93, 1.70)	0.887	−1.39 (−2.87, 0.11)	0.070

SF, serum ferritin; TC, total cholesterol; LDL-C, low-density lipoprotein cholesterol; TG, triglyceride; HDL-C, high-density lipoprotein cholesterol. Linear regression models with adjustment for age, race, six-month time period, educational level, marital status, BMI, hypertension, smoking status, alcohol status, daily total energy intake, and total daily fat intake. ^1^ Blood lipid (TC, LDL-C, TG, and HDL-C) concentrations and SF measures were ln-transformed in models. Results presented as percent differences in blood lipid concentrations in relation to a 1-unit increase in SF concentrations. For every 1-unit increase in serum ferritin value, the mean value of blood lipids increased by *β*%. Percent differences = [e^(*β*)^ − 1] × 100.

**Table 3 nutrients-15-01878-t003:** ORs (95% CIs) for four types of dyslipidemia according to daily total iron intake quartile categories in males and females.

Variables			Category of Daily Total Iron Intake, μg/L; OR (95% CI)				*p* Trend
Gender (*n* = 2525)	Dyslipidemia Type	Model	Q1	Q2	Q3	Q4	
Males (*n* = 1307)							
	High TC	Crude	1.00 (ref.)	1.27 (0.73, 2.21)	1.14 (0.65, 2.00)	1.00 (0.56, 1.78)	0.797
		Adjusted	1.00 (ref.)	1.38 (0.76, 2.51)	1.43 (0.73, 2.81)	1.51 (0.68, 3.37)	0.691
	High LDL-C	Crude	1.00 (ref.)	1.07 (0.64, 1.79)	0.80 (0.46, 1.37)	0.83 (0.48, 1.42)	0.644
		Adjusted	1.00 (ref.)	1.25 (0.72, 2.18)	1.08 (0.56, 2.08)	1.39 (0.65, 2.96)	0.763
	High TG	Crude	1.00 (ref.)	1.19 (0.67, 2.10)	1.62 (0.95, 2.79)	1.18 (0.67, 2.09)	0.336
		Adjusted	1.00 (ref.)	1.27 (0.69, 2.34)	1.71 (0.91, 3.22)	1.35 (0.64, 2.86)	0.395
	Low HDL-C	Crude	1.00 (ref.)	1.17 (0.81, 1.70)	0.89 (0.62, 1.27)	1.09 (0.76, 1.58)	0.475
		Adjusted	1.00 (ref.)	1.03 (0.68, 1.55)	0.86 (0.55, 1.32)	0.97 (0.59, 1.61)	0.808
Females (*n* = 1218)							
	High TC	Crude	1.00 (ref.)	0.94 (0.58, 1.53)	0.98 (0.6, 1.58)	0.63 (0.37, 1.06)	0.261
		Adjusted	1.00 (ref.)	0.97 (0.57, 1.65)	0.93 (0.52, 1.67)	0.65 (0.32, 1.31)	0.560
	High LDL-C	Crude	1.00 (ref.)	0.87 (0.52, 1.46)	0.94 (0.57, 1.57)	0.68 (0.39, 1.17)	0.526
		Adjusted	1.00 (ref.)	0.98 (0.56, 1.71)	1.10 (0.60, 2.03)	0.87 (0.42, 1.77)	0.884
	High TG	Crude	1.00 (ref.)	1.39 (0.63, 3.06)	2.60 (1.27, 5.34)	1.37 (0.62, 3.03)	0.037
		Adjusted	1.00 (ref.)	1.56 (0.67, 3.64)	3.16 (1.38, 7.23)	1.53 (0.57, 4.12)	0.020
	Low HDL-C	Crude	1.00 (ref.)	1.21 (0.88, 1.66)	1.17 (0.85, 1.61)	0.98 (0.72, 1.35)	0.454
		Adjusted	1.00 (ref.)	1.11 (0.78, 1.58)	0.99 (0.68, 1.46)	0.81 (0.53, 1.25)	0.427

OR, odds ratio; CI, confidential intervals; Q1: 1st quartile, Q2: 2nd quartile, Q3: 3rd quartile, Q4: 4th quartile; TC, total cholesterol; LDL-C, low-density lipoprotein cholesterol; TG, triglyceride; HDL-C, high-density lipoprotein cholesterol; OR and 95% CI were obtained by multivariable logistic regression analysis (crude models). The adjusted models were adjusted for age, race, six-month time period, educational level, marital status, BMI, hypertension, smoking status, alcohol status, total daily energy intake, and total daily fat intake.

## Data Availability

The dataset used for our study is publicly available from the National Center for Health Statistics of the Centers for Disease Control and Prevention: https://www.cdc.gov/nchs/nhanes/index.htm (accessed on 1 February 2023).

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
