# Peer review of "Relationship between Serum Ferritin Level and Dyslipidemia in US Adults Based on Data from the National Health and Nutrition Examination Surveys 2017 to 2020"

_nutrients, 2023, doi:10.3390/nu15081878_

Round 1
Reviewer 1 Report
The manuscript entitled "Relationship between serum ferritin level and dyslipidemia in the US adults: based on the data from the National Health and Nutrition Examination Surveys 2017 to 2020" presents an association study between the serum levels of ferritin (SF) and dyslipidemia in adults using data extracted from NHANES. The aim of the manuscript is very interesting, nevertheless it must be improved.
Following are described my comments:
1. Globally, the manuscript lacks organization in terms of formatting, which makes it difficult to read. Furthermore, the images have low quality. They are not perceptible. These issues must be improved.
2. The sentence described in introduction: “Studies found that obesity, metabolic syndrome, atherosclerosis, diabetes mellitus, dyslipidemia and cancer SF are also positively associated with SF[18-22].” is very puzzling and should be improved.
3. Abbreviations must be avoided in the figures and in legends. When used, they should, at least, be explained in the legend.
4. in the methods section, specifically in sub-section “2.5. Statistical Analyses”, the sentence: “The 1st quartile(Q1) and 4th quartile(Q2) represented the lowest and highest ferritin levels, respectively.” has a mistake.
5. In the sub-section “2.4. Assessment of covariates”, according to the included information, authors have defined several groups. Which were the criteria employed to defined the aged and race/ethnicity groups? At least, the definition of these groups should be taken into consideration. A discussion about these groups must be included. Furthermore, in the literature there is at least one reference -doi:10.1016/j.ccl.2015.01.005- correlating dyslipidemia and ethnic populations, which must be emphasized in this manuscript.
6. Another question is related with the definition of the 4 quartiles for SF by gender. Which are the number of individuals associated with each quartile? Still related with the SF levels, they are completely distinct between female and male. Is it normal? Or is it a drawback of the study. Since the SF levels are central in the aim of this manuscript, this aspect should be taken into consideration. Furthermore, the authors also describe 4 types of dyslipidemia, but they did not explore the definition of these types or the difference between male and female. These aspects must be improved in the manuscript.
Reviewer 2 Report
The Authors used data from the National Health and Nutrition Examination
Surveys (NHANES) to investigate the dyslipidemia relationships with serum ferritin (SF). Authors conclude that dietary iron intake is associated with high dyslipidemia. The study is of interest, however, I see an over-interpretation of the results: high dietary iron intake -> high SF -> dyslipidemia.
I would find more clear explanations in the manuscript concerning the following major points.
Are explicit data about daily dietary iron intake present in the study from the NHANES data set? I understand that SF is the parameter used in the study, without data about the daily dietary intake. The authors should explain if SF level is exclusively determined by daily dietary iron intake, or the values should be due to other conditions (as an example, altered pathways due to dyslipidemia ?). In other word, could the dyslipidemia state be the reason for high level of SF ? An important methodological aspect concerns paragraph 2.2: it is not clear if the serum specimens have been analyzed within this work or, alternatively, the authors used values already included into the NHANES dataset.
The following minor points are related to the quality of the manuscript presentation:
- Check under Introduction "cancer SF"
- Table 1. First column is not clearly labeled. age, race, education level, marital status, drinking status, etc. are main characteristics, shold be evidenced in respect of the other lines, and with similar formatting features (i.e., why age is in a underlined cell, and the other not ?)
- The term "race" is object of careful cheking for its use. Please check the terms used and follow the guideline of the article: https://jamanetwork.com/journals/jama/fullarticle/2776936
- Table 2. Numbers in parentesis are not explained in clear manner.
- Figure 2 and 3 appear of very low quality. The origin of the images is unclear.
Round 2
Reviewer 2 Report
I reccommend a few more revisions under proof correction, if not applied at editorial level:
- names of the authors do not appear with appropriate font size
- line 283 on page 11: storge (to be corrected)
- check the size of figures to obtain better readability
